# A Pilot Study of the Clinical Frailty Scale to Predict Frailty Transition and Readmission in Older Patients in Vietnam

**DOI:** 10.3390/ijerph17051582

**Published:** 2020-02-29

**Authors:** Tan Van Nguyen, Thuy Thanh Ly, Tu Ngoc Nguyen

**Affiliations:** 1Department of Geriatrics & Gerontology, University of Medicine and Pharmacy, Ho Chi Minh City, Vietnam; udoctor2014@gmail.com; 2Department of Interventional Cardiology, Thong Nhat Hospital, Ho Chi Minh City, Vietnam; 3Geriatrics Department, Nguyen Trai Hospital, Ho Chi Minh City, Vietnam; 4Westmead Applied Research Centre, Faculty of Medicine and Health, The University of Sydney, Sydney, NSW 2145, Australia; ngoc.tu.nguyen@sydney.edu.au

**Keywords:** frailty, Clinical Frailty Scale, frailty transition, older patients, Vietnam

## Abstract

Background. The Clinical Frailty Scale (CFS) is gaining increasing acceptance due to its simplicity and applicability. Aims. This pilot study aims to examine the role of CFS in identifying the prevalence of frailty, frailty transition, and the impact of frailty on readmission after discharge in older hospitalized patients. Methods. Patients aged ≥60 admitted to the geriatric ward of a hospital in Vietnam were recruited from 9/2018–3/2019 and followed for three months. Frailty was assessed before discharge and after three months, using the CFS (robust: score 1–2, pre-frail: 3–4, and frail: ≥5). Multivariate logistic regression was applied to investigate the associated factors of frailty transition and the impact of frailty on readmission. Results. There were 364 participants, mean age 74.9, 58.2% female. At discharge, 4 were robust, 160 pre-frail, 200 frail. Among the 160 pre-frail participants at discharge, 124 (77.5%) remained pre-frail, and 36 (22.5%) became frail after 3 months. Age (adjusted OR1.09, 95% CI 1.03–1.16), number of chronic diseases (adjusted OR 1.37, 95% CI 1.03–1.82), and polypharmacy at discharge (adjusted OR 3.68, 95% CI 1.15–11.76) were significant predictors for frailty after 3 months. A frailty status at discharge was significantly associated with increased risk of readmission (adjusted OR2.87, 95% CI 1.71–4.82). Conclusions. Frailty was present in half of the participants and associated with increased risk of readmission. This study suggests further studies to explore the use of the CFS via phone calls for monitoring patients’ frailty status after discharge, which may be helpful for older patients living in rural and remote areas.

## 1. Introduction

Frailty, defined as a state of decreased physiological reserve and increased vulnerability to stressor events, can increase the risk of having adverse outcomes in older hospitalized patients [1,2]. Frailty occurs as a result of multiple physical, social, and environmental factors. Although the concept of frailty has been emerging in ageing research and geriatric medicine for many years, there is little consensus on the assessment and diagnosis of frailty, especially in clinical settings [1,2]. The most commonly used definitions revolve around deficit accumulation model (the Frailty Index) and the frailty phenotype (or Fried’s frailty criteria) [1,3,4]. These two frailty definitions are powerful predictors of adverse health outcomes in older people but they are time-consuming and require many physical measurements, which may not be feasible for older hospitalized patients with acute illnesses, and particularly in resource-limited settings [2]. In recent years, the Clinical Frailty Scale (CFS) was developed and validated to provide clinicians with a more feasible approach to detect frailty in older hospitalized patients [5]. It is a well-validated 9-point global assessment tool that not only focuses on the physical aspect but also other clinical domains such as comorbidities and energy level. This scale has been recommended as one of the frailty screening tools by the Asia-Pacific Clinical Practice Guidelines for the Management of Frailty [2]. Its efficiency, reliability, and validity meet the needs of acute care settings. CFS scores can predict readmission and other adverse health outcomes [2]. The CFS score ranges from 1–9 and can be classified into robust (score 1–2), pre-frail (3–4), and frail (≥5) [5,6].

The prevalence of frailty in community-dwelling older adults ranged from 4% to 10% in studies in Western countries, and was approximately 3.5–27% in the Asia-Pacific region [2]. In Vietnam, frailty is of concern given the rapidly ageing population. There is thus a need to identify frailty and to provide appropriate care for the frail seniors. The evidence of frailty in Vietnam, although limited, showed that frailty was associated with negative outcomes in older people [7,8,9,10]. Frailty research in Vietnam has evolved in the past couple of years, however there has been no study examining the use of the CFS in identifying frailty in older hospitalized patients. Therefore, in this pilot study, we aim to apply the CFS in identifying the prevalence of frailty, frailty transition, and the impact of frailty on readmission after discharge in older hospitalized patients.

## 2. Methods

### 2.1. Participants

A prospective cohort study was conducted in patients admitted to the geriatric ward of Nguyen Trai Hospital, Ho Chi Minh City, Vietnam from 9/2018 to 3/2019. All patients aged 60 years or older were eligible for the study. Exclusion criteria include: (1) unable to understand and answer the study questions (including severe dementia), (2) having cancer with life expectancy of less than 3 months, (3) unable to obtain consent.

### 2.2. Sample Size Calculation

The sample size was calculated based on the first aim of the study (to investigate the prevalence of frailty using CFS). We used a single population proportion formula: n = Z^2^
_1−α/2_ ∗ [p∗(1 − p)/d^2^], with n = the required sample size, Z_1−α/2_ = 1.96 (with α = 0.05 and 95% confidence interval), p = prevalence of frailty in older patients, and d = precision (assumed as 0.05). According to a previous study in the north of Vietnam, the prevalence of frailty in older patients admitted to the National Geriatric Hospital was 31.9–35.4% [7]. Therefore, the sample size for this study is calculated to be around 350 participants.

### 2.3. Data Collection

Data were collected from patient interviews and from medical records, using a predefined data collection sheet. Information obtained from medical records included: socio-demographic characteristics, height, weight, medical history, comorbidities, diagnosis, and medications at discharge. Polypharmacy was judged based on the prescriptions at discharge (defined as five or more prescribed medications). Frailty was assessed prior to discharge by one of the study investigators (TTL)—a geriatrician practicing at Nguyen Trai Hospital—using the CFS.

The CFS divides participants into 9 categories from very fit (CFS = 1) to terminally ill (CFS = 9) [5,6,11], as follows:-CFS 1 (very fit): People who are robust, active, energetic, and motivated. These people commonly exercise regularly. They are among the fittest for their age.-CFS 2 (well): People who have no active disease symptoms but are less fit than category 1. Often, they exercise or are very active occasionally.-CFS 3 (managing well): People whose medical problems are well controlled, but are not regularly active beyond routine walking.-CFS 4 (vulnerable): While not dependent on others for daily help, often symptoms limit activities. They often complain of being “slowed up”, and/or being tired during the day.-CFS 5 (mildly frail): These people often have more evident slowing, with limited dependence on others for instrumental activities of daily living.-CFS 6 (moderately frail): People need help with all outside activities and with keeping house. They often have problems with climbing stairs and need help with bathing, and may need assistance with dressing.-CFS 7 (severely frail): Completely dependent for personal care. However, they seem stable and not at high risk of dying within 6 months.-CFS 8 (very severely frail): Completely dependent, approaching the end of life. Typically, they could not recover from a minor illness.-CFS 9 (terminally ill): Approaching the end of life. This category applies to people with a life expectancy less than 6 months, who are not otherwise evidently frail.

All participants were followed for 3 months. The same investigator (TTL) conducted phone calls to the participants at the end of the third month after discharge to obtain information about readmission and to assess frailty using the CFS. Frailty transition was defined as the transition from robust to pre-frail status, or from pre-frail to frail status. Readmission information was achieved by asking the participants and by checking in the medical record system of the hospital using the participants’ registration numbers. Readmission was recorded as a binary variable (yes/no).

### 2.4. Statistical Analysis

Analysis of the data was performed using SPSS for Windows 24.0 (IBM Corp., Armonk, NY, USA). Continuous variables are presented as means ± standard deviation, and categorical variables as frequencies and percentages. Participants were classified into three groups according to their CFS scores: robust (score 1–2), pre-frail (3–4), and frail (≥ 5), as recommended in previous studies [5,6]. Comparisons between frail and robust/pre-frail participants were conducted using the Chi-square test or Fisher’s exact test for categorical variables and Student’s t-test or Mann–Whitney test for continuous variables. 

To identify the factors independently associated with frailty transition after 3 months, multivariable logistic regression analysis was applied. First, univariate logistic regression was performed on all the potential associated factors for frailty among participants who were pre-frail at discharge. Then variables that had a *p*-value <0.20 on univariate analysis were selected for multivariate analysis. A backward elimination method was applied so that the final model retained only those variables significant at *p* < 0.05. 

To investigate the impact of frailty at discharge on readmission, first we conducted univariate logistic regression of frailty on readmission. The relationship between frailty at discharge and readmission were then examined by multivariate logistic regression, adjusted to those variables that had a *p*-value <0.05 on univariate analyses. 

All variables were examined for interaction and multicollinearity. Results were presented as odds ratios and 95% confidence intervals.

### 2.5. Ethical Approval

The study protocol was approved by the ethics committees of the University of Medicine and Pharmacy, Ho Chi Minh City, Vietnam. All procedures followed were in accordance with the ethical standards of the responsible committee on human experimentation and with the Helsinki Declaration of 1964, as revised in 2013. Informed consent was obtained from all participants for being included in the study. Participants could withdraw anytime without it affecting their current treatment. Their information was kept confidential and used only for research purposes.

## 3. Results

There were 364 participants. They had a mean age of 74.9 years and 58.2% were female. The distribution of the Clinical Frailty Scale categories is presented in Figure 1. The two most frequent CFS categories were 4 and 5. At discharge, among 364 participants, 4 were robust, 160 pre-frail, and 200 frail. The prevalence of frailty was 54.9%. Compared to the robust/pre-frail participants, the frail were significantly older, more likely to have a low education level, and to be alone. Frail participants also had higher prevalence of underweight and polypharmacy at discharge. The mean number of chronic diseases was also higher in the frail group (Table 1).

### 3.1. Frailty Transition after 3 Months

The number of robust participants at discharge (*n* = 4) was too small for frailty transition analysis.

Among 160 pre-frail participants at discharge, 124 (77.5%) remained pre-frail, and 36 (22.5%) became frail (Figure 2). Table 2 presents the univariate logistic regression analyses of potential factors associated with frailty transition after 3 months among these 160 participants. On multivariate logistic regression, age (adjusted OR 1.09, 95% CI 1.03–1.16 for each additional year), number of chronic diseases (adjusted OR 1.37, 95% CI 1.03–1.82 for each additional chronic disease), and polypharmacy (having at least 5 medications) at discharge (adjusted OR 3.68, 95% CI 1.15–11.76) were significant predictors for frailty (Table 3).

### 3.2. The Impact of Frailty at Discharge on Readmission 

During 3 months of follow-up, the proportion of readmission was 26.1% (95/364) in all participants, 15.2% in the robust/pre-frail and 35.0% in the frail (*p* < 0.001).

Univariate logistic regression of predictive factors for readmission is presented in Table 4. On univariate logistic regression, frailty was associated with a 3-fold increased risk of readmission (unadjusted OR 2.99, 95% CI 1.79–5.01). The impact of frailty on readmission was still significant after adjustment for the number of chronic diseases (adjusted OR 2.87, 95% CI 1.71–4.82, *p* < 0.001). 

## 4. Discussion

In this study on 364 participants admitted to a geriatric ward of a general hospital in Vietnam, we found that frailty defined by the CFS was present in 54.9% of the participants. Older age, increased number of chronic diseases, and polypharmacy at discharge were significant predictors for frailty transition after 3 months. A frailty status at discharge significantly predicted readmission during the 3 months follow up.

Our findings are compatible with some other studies in the region. In the Singapore Longitudinal Ageing Study 2 on 1297 older community-dwellers, older age was one of the significant predictors for frailty transition (defined by the frailty phenotype) after 3–5 years [12]. In another study in Indonesia, older age, low quality of life, and slow gait speed were prognostic factors for frailty transition (defined by the Frailty Index) after 12 months [13]. In a study on 696 older people in Australia, chronological age and multimorbidity were associated with frailty transition during 4.5 years of follow up (with frailty defined by the frailty phenotype and by the Frailty Index) [14]. People with multiple chronic diseases have a higher risk of physical frailty due to reduced physical activity, change in diet, and reduced protein intake, and complex interactions between diseases and medications [15]. As the number of chronic diseases and polypharmacy at discharge were significantly associated with a transition into frailty after 3 months, our study suggest that frailty screening should be performed in older hospitalized patients, especially in patients with multiple chronic diseases in Vietnam. The prevalence of polypharmacy in older people in Vietnam was very high, as reported in a previous study (63% in the frail and 57% in the non-frail) [10]. There should be strategies to conduct regular medication reviews and to reduce polypharmacy in older patients. More effort is needed to raise awareness about polypharmacy among older patients and health professionals in Vietnam. 

Our findings also suggest that the CFS may be feasible for monitoring frailty status in older patients after discharge. The CFS is gaining popularity due to its simplicity and applicability. This scale can also allow health professionals who are not specialized in geriatrics to conduct accurate assessments. In a study on 179 older patients in Australia in 2013, the CFS was reported to be acceptable and feasible for junior medical staff to identify patients’ baseline frailty status in the acute general medical setting, using information obtained on routine clinical assessment [16]. In a recent study conducted on 184 older patients at the Acute Geriatric Unit of a University Hospital in Spain, there was a strong correlation of the CFS with a frailty index (*r* = 0.706, *p* < 0.001) [17]. In another study on 71 older patients with pelvic floor conditions in the United States, the CFS also had good agreement with the frailty index in identifying frailty [18]. In Vietnam, over the past 5 years, there have been several studies on frailty in older people in the acute care settings and in the community. The Reported Edmonton Frail Scale and Fried’s frailty phenotype were applied in these studies. In a study on 461 hospitalized older patients admitted to a geriatric hospital, the prevalence of frailty was 31.9% according to the Reported Edmonton Frail Scale and 35.4% according to Fried’s frailty phenotype, and frailty was associated with increased mortality during 6 months of follow up [7]. In a study on 523 older adults in the northern rural areas of Vietnam, the prevalence of frailty defined by Fried’s frailty phenotype was 21.7% and was associated with reduced health related quality of life [8]. In another study on 324 older patients admitted to cardiology ward of a tertiary hospital in Vietnam, frailty–defined by the Reported Edmonton Frail Scale–was present in 48.1% and associated with increased adverse outcomes such as hospital-acquired pneumonia, mortality, and readmission [9]. However, there has been no study using the CFS in older patients in Vietnam. While the Reported Edmonton Frail Scale [19] and Fried’s frailty phenotype [3] require some face-to-face assessment, the CFS assessment can be conducted via phone calls and hence is helpful for older patients who live in rural and remote areas in Vietnam. The identification of frailty in older patients can help develop frailty-tailored treatments, such as early mobility, nutritional assessment and management, and medication review to reduce potential adverse events.

### Strength and Limitations

This is the first study to examine the role of the CFS in identifying the prevalence of frailty, frailty transition and the impact of frailty on readmission after discharge in older hospitalized patients in Vietnam. This study contained a sample of older patients with high quality detailed clinical information. However, this study was conducted at a geriatric ward of a single hospital in Vietnam, which may not be representative for all older patients in Vietnam. In addition, the CFS was conducted via phone calls and by the same investigator who performed the CFS assessment prior to discharge. This could cause bias in scoring. Further studies are needed to confirm the validity of frailty assessment over the phone using the CFS. Therefore, results should be cautiously interpreted and should not be generalized to all older patients.

## 5. Conclusions

In this study, frailty was present in half of the participants and associated with increased risk of readmission. Older age, multiple chronic diseases, and polypharmacy increased the risk of transition into frailty. This study suggests further studies to explore the use of CFS via phone calls for monitoring patients’ frailty status after discharge, which may be helpful for older patients living in rural and remote areas.

## Figures and Tables

**Figure 1 ijerph-17-01582-f001:**
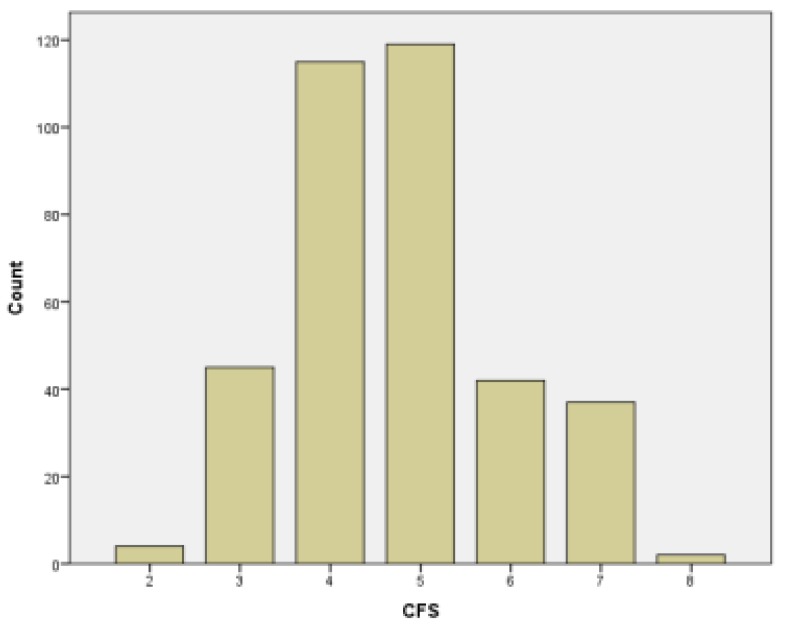
Distribution of the Clinical Frailty Scale categories.

**Figure 2 ijerph-17-01582-f002:**
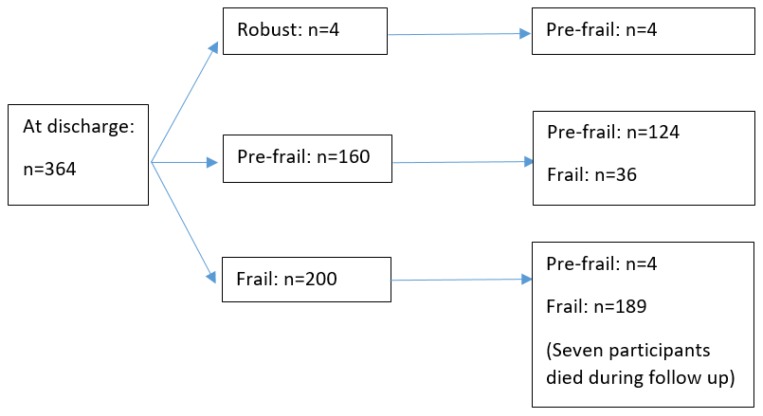
Frailty transition after 3 months.

**Table 1 ijerph-17-01582-t001:** General characteristics by frailty status.

Variables	All (*n* = 364)	Robust/Pre-frail (*n* = 164)	Frail (*n* = 200)	*p*
Age, years	74.9 ± 9.4	69.4 ± 6.8	79.4 ± 8.7	<0.001
Female	212 (58.2)	100 (61.0)	112 (56.0)	0.338
Education level:				
Low (illiterate or primary school)	122 (33.5)	41 (25.0)	81 (40.5)	0.007
Intermediate (secondary or high school)	171 (47.0)	85 (51.8)	86 (43.0)	
High (higher than high school)	71 (19.5)	38 (23.2)	33 (16.5)	
Marital status:				
Married	203 (55.8)	104 (63.4)	99 (49.5)	0.018
Widowed	132 (36.3)	45 (27.4)	87 (43.5)	
Never married	18 (4.9)	9 (5.5)	9 (4.5)	
Divorced/separated	11 (3.0)	6 (3.7)	5 (2.5)	
Body mass index				
Underweight (<18.5)	41 (11.4)	8 (4.9)	33 (16.8)	0.001
Normal (18.5 ≤ 25.0)	236 (65.7)	110 (67.9)	126 (64.0)	
Overweight (≥25.0)	82 (22.8)	44 (27.2)	38 (19.3)	
Number of chronic diseases	3.65 ± 1.59	3.47 ± 1.65	3.80 ± 1.53	0.048
Having ≥2 chronic diseases	324 (89.0)	142 (86.6)	182 (91.0)	0.180
Polypharmacy at discharge	264 (72.5)	110 (67.1)	154 (77.0)	0.035
Main diagnosis at discharge:				
Hypertension	172 (47.3)	85 (51.8)	87 (43.5)	0.146
Infection	78 (21.4)	24 (14.6)	54 (27.0)	
Gastrointestinal disorder	20 (5.5)	11 (6.7)	9 (4.5)	
Musculoskeletal pain	16 (4.4)	8 (4.9)	8 (4.0)	
Chronic lung diseases	14 (3.8)	5 (3.0)	9 (4.5)	
Ischemic heart disease	9 (2.5)	5 (3.0)	4 (2.0)	
Stroke	9 (2.5)	3 (1.8)	6 (3.0)	
Other	46 (12.6)	23 (14.0)	23 (11.5)	

Continuous data are presented as mean ± standard deviation. Categorical data are shown as *n* (%).

**Table 2 ijerph-17-01582-t002:** Univariate logistic regression of potential associated factors for frailty transition among participants who were pre-frail before discharge (*n* = 160).

Variables	Unadjusted Odds Ratios for Frailty (95% CI)	*p*
Age	1.12 (1.06–1.18)	<0.001
Female	1.20 (0.55–2.58)	0.649
Underweight	0.48 (0.06–4.02)	0.496
Overweight	0.57 (0.23–1.41)	0.223
Low education	0.67 (0.27–1.66)	0.384
Being alone (single/divorced/widow)	1.75 (0.82–3.71)	0.146
Polypharmacy at discharge	5.23 (1.74–15.70)	0.003
Number of chronic diseases	1.60 (1.22–2.09)	0.001
Main reasons for admission:		
Hypertension	1.05 (0.50–2.20)	0.902
Cardiovascular disease (ischemic heart disease, stroke, heart failure)	2.46 (0.65–9.24)	0.183
Chronic lung disease	5.55 (0.89–34.57)	0.067
Skeletomuscular pain	3.75 (0.89–15.82)	0.072

**Table 3 ijerph-17-01582-t003:** Predictive factors for transition to frailty in multivariable logistic regression.

Variables	Adjusted Odds Ratios for Frailty (95% CI)	*p*
Age	1.09 (1.03–1.16)	0.003
Number of chronic diseases	1.37 (1.03–1.82)	0.030
Polypharmacy at discharge	3.68 (1.15–11.76)	0.028

Variables entered in step 1: age, being alone, polypharmacy at discharge, number of chronic diseases, main reason of admission due to cardiovascular diseases, main reason of admission due to chronic lung disease, main reason of admission due to skeletomuscular pain.

**Table 4 ijerph-17-01582-t004:** Univariate logistic regression of predictive factors for readmission.

Variables	Unadjusted Odds Ratios for Readmission (95% CI)	*p*
Frailty at discharge	2.99 (1.79–5.01)	<0.001
Age	1.02 (0.99–1.05)	0.091
Female	0.93 (0.58–1.49)	0.748
Underweight	1.75 (0.88–3.46)	0.111
Overweight	1.04 (0.60–1.80)	0.890
Low education	1.08 (0.66–1.76)	0.769
Being alone (single/divorced/widow)	1.06 (0.66–1.69)	0.814
Polypharmacy at discharge	1.08 (0.64–1.84)	0.769
Number of chronic diseases	1.24 (1.07–1.45)	0.006

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
