# Peer review of "A Pilot Study of the Clinical Frailty Scale to Predict Frailty Transition and Readmission in Older Patients in Vietnam"

_ijerph, 2020, doi:10.3390/ijerph17051582_

Round 1

Reviewer 1 Report

This study examines the use of the Clinical Frailty Scale (CFS) in assessing the level of frailty among older adults in Vietnam and to examine the association of the prevalence of frailty (via CFS scores) with frailty transition  and readmission. While the study does have some significant limitations (e.g., it being conducted at 1 location with a single investigator measuring frailty and assigning CFS scores), I think as a pilot study it presents strong evidence to support the notion of using the CFS in clinical settings in Vietnam. Below are a few minor critiques:

  • I‘m somewhat confused as to how the 9-point scale of the CFS was reduced down to the categories of robust, pre-fail, and frail by the investigators. The abstract implies that scores of 1-2 were recoded as robust, 3-4 to pre-frail, and 5 and over to frail – however, this recoding is not presented in the actual body of the manuscript (or if it is, I missed it). This should be more clearly illustrated in the methods (and a reasoning/justification behind the recoding must also be given).
  • It appears that certain items are either missing or were “cut off” in Figure 2 (possibly when pasting the figure into the document?). A new figure should replace it with all the necessary information readable, because as it is now it may confuse the reader.

Author Response

Thank you for the time spent reviewing our manuscript and for the useful comments. Please see our response as follows:

This study examines the use of the Clinical Frailty Scale (CFS) in assessing the level of frailty among older adults in Vietnam and to examine the association of the prevalence of frailty (via CFS scores) with frailty transition and readmission. While the study does have some significant limitations (e.g., it being conducted at 1 location with a single investigator measuring frailty and assigning CFS scores), I think as a pilot study it presents strong evidence to support the notion of using the CFS in clinical settings in Vietnam. Below are a few minor critiques:

I‘m somewhat confused as to how the 9-point scale of the CFS was reduced down to the categories of robust, pre-fail, and frail by the investigators. The abstract implies that scores of 1-2 were recoded as robust, 3-4 to pre-frail, and 5 and over to frail – however, this recoding is not presented in the actual body of the manuscript (or if it is, I missed it). This should be more clearly illustrated in the methods (and a reasoning/justification behind the recoding must also be given).

Response: The grouping of the CFS score was presented in the Introduction of the manuscript. To make it clearer, we have added the description of cut-offs and justification in the Methods (Please see Page 6, Statistical Analysis).

It appears that certain items are either missing or were “cut off” in Figure 2 (possibly when pasting the figure into the document?). A new figure should replace it with all the necessary information readable, because as it is now it may confuse the reader.

Response: Figure 2 might have been changed when the file was converted to pdf. We have changed the format of Figure 2 to avoid this (Please see Figure 2)

Reviewer 2 Report

This pilot study investigates the utility of the Clinical Frailty Scale (CFS) in identifying the prevalence of frailty, as well as frailty transitions and the impact of frailty on readmissions in 364 patients aged over 60 years admitted to a geriatrics ward in a single hospital in Vietnam. The authors report that 55% were frail (200/364), and among the pre-frail, 22.5% had become frail after 3 months follow-up. The predictors of frailty after 3 months were age, chronic disease and polypharmacy. Frailty at discharge was also found to be associated with an increased risk of readmission.

Overall, these are interesting findings from a region where data are sparse. In addition, the ease of use of the CFS means that it is garnering increasing interest both in research and in clinical practice. It does not require a battery of complex geriatric assessments, and yet is accurate and meaningful.

I have a few comments for the authors’ consideration (in no particular order of importance):

Firstly, did the authors verify the sensitivity to change (or test-retest value) of the CFS instrument, as they performed the two assessments at 3 months’ interval. Do the authors think that a change (notably a transition towards increasing frailty) at 3 months is a definitive change, and what about its relation to the acute event that caused the initial hospital admission? The authors do not investigate the backward transitions (pre-frail who become robust, or frail patients who move “up” the scale to pre-frail status). Perhaps some of the transitions were a response to the acute event that may be resolved after some weeks or months of convalescence. I think a few lines about this issue in the discussion would be useful.

Regarding the re-admissions, do the authors have any data about the motives for re-admission? Indeed, it would have been interesting to know the proportion of re-admission for chronic disease, for example, or re-admission for the same cause as the initial admission. Furthermore, were these re-admissions to the geriatrics ward, or to any hospital department? If this information is not available, it should be noted in the limitations of the study.

Results – in presenting the odds ratios (OR) from the multivariate regression, the authors must specify the increment associated with the OR. For age, for example, the OR is 1.09, indicating an increase of 9% per increment, but what is this increment? Per additional year? Or per 5-years? Please specify. The same applies for chronic diseases (per additional comorbidity?) and polypharmacy (per additional drug? Or versus no polypharmacy?).

In Figure 2, the sum of the groups in the box in the bottom right hand corner (transitions of the frail group) is not 200 – please verify. Also, some of the numbers are not visible on the figure (the number of pre-frail in the middle column, and the number overall in the first column is partially cut off). I imagine that this is just a formatting problem.

Finally, the authors document here the utility of the CFS for detecting frailty and patients at risk of readmission but what are the perspectives for these patients in the Vietnamese context, once their risk has been identified? The major utility of screening is to orient potentially at-risk individuals towards specific management, but what can be proposed to, or done for these patients? I think a few lines about the perspectives for management would be useful.

Overall, there are a number of minor grammar/spelling errors that merit correction – below are a few examples, but I would advise thorough proofreading of the text.

  • Line 63, were eligible (not are eligible)
  • Line 172, “after adjustment for the number of chronic diseases” (not adjusted to)
  • Line 176, was present in 54.9% (not presented)
  • Line 197, the CFS is gaining popularity (not gaining ITS popularity)
  • Line 200, junior medical what? There seems to be a word missing (staff?)
  • Line 204, the CFS also had good agreement (not have)
  • Line 228, should not be generalised to all older patients

Author Response

Thank you for the time spent reviewing our manuscript and for the useful comments. We feel that the suggestions have strengthened the manuscript and have tried to address as many of the suggestions as possible as detailed below.

This pilot study investigates the utility of the Clinical Frailty Scale (CFS) in identifying the prevalence of frailty, as well as frailty transitions and the impact of frailty on readmissions in 364 patients aged over 60 years admitted to a geriatrics ward in a single hospital in Vietnam. The authors report that 55% were frail (200/364), and among the pre-frail, 22.5% had become frail after 3 months follow-up. The predictors of frailty after 3 months were age, chronic disease and polypharmacy. Frailty at discharge was also found to be associated with an increased risk of readmission.

Overall, these are interesting findings from a region where data are sparse. In addition, the ease of use of the CFS means that it is garnering increasing interest both in research and in clinical practice. It does not require a battery of complex geriatric assessments, and yet is accurate and meaningful.

I have a few comments for the authors’ consideration (in no particular order of importance):

Firstly, did the authors verify the sensitivity to change (or test-retest value) of the CFS instrument, as they performed the two assessments at 3 months’ interval. Do the authors think that a change (notably a transition towards increasing frailty) at 3 months is a definitive change, and what about its relation to the acute event that caused the initial hospital admission? The authors do not investigate the backward transitions (pre-frail who become robust, or frail patients who move “up” the scale to pre-frail status). Perhaps some of the transitions were a response to the acute event that may be resolved after some weeks or months of convalescence. I think a few lines about this issue in the discussion would be useful.

Response: Thank you for your suggestion. We have added the reasons for admission in logistic regression models to predict frailty transition. The final model remained unchanged. (Please see Table 2 and Table 3). We think that the role of acute events were not as significant as factors such as advanced age, polypharmacy and the accumulated number of chronic diseases.

There were 4 frail participants at discharge that became pre-frail after 3 months. However the number is too small for further analysis.

Regarding the re-admissions, do the authors have any data about the motives for re-admission? Indeed, it would have been interesting to know the proportion of re-admission for chronic disease, for example, or re-admission for the same cause as the initial admission. Furthermore, were these re-admissions to the geriatrics ward, or to any hospital department? If this information is not available, it should be noted in the limitations of the study.

Response: Unfortunately we did not record details of readmission.

Results – in presenting the odds ratios (OR) from the multivariate regression, the authors must specify the increment associated with the OR. For age, for example, the OR is 1.09, indicating an increase of 9% per increment, but what is this increment? Per additional year? Or per 5-years? Please specify. The same applies for chronic diseases (per additional comorbidity?) and polypharmacy (per additional drug? Or versus no polypharmacy?).

Response: Thank you for your suggestion. We have made some modification as suggested (Page 6)

On multivariate logistic regression, age (adjusted OR 1.09, 95%CI 1.03 – 1.16 for each additional year), number of chronic diseases (adjusted OR 1.37, 95%CI 1.03 – 1.82 for each additional chronic disease) and polypharmacy (having at least 5 medications) at discharge (adjusted OR 3.68, 95%CI 1.15 – 11.76) were significant predictors for frailty. (Table 3)

In Figure 2, the sum of the groups in the box in the bottom right hand corner (transitions of the frail group) is not 200 – please verify. Also, some of the numbers are not visible on the figure (the number of pre-frail in the middle column, and the number overall in the first column is partially cut off). I imagine that this is just a formatting problem.

Response: We are sorry that there was a mistake with Figure 2. The number of frail participants who died during follow up was 7, not 9. We have made changes (please see Figure 2)

Finally, the authors document here the utility of the CFS for detecting frailty and patients at risk of readmission but what are the perspectives for these patients in the Vietnamese context, once their risk has been identified? The major utility of screening is to orient potentially at-risk individuals towards specific management, but what can be proposed to, or done for these patients? I think a few lines about the perspectives for management would be useful.

Response: Thank you for your suggestion. We have added more discussion as follows (Please see Page 10)

“The identification of frailty in older patients can help develop frailty-tailored treatments, such as early mobility, nutritional assessment and management, and medication review to reduce potential adverse events.”

Overall, there are a number of minor grammar/spelling errors that merit correction – below are a few examples, but I would advise thorough proofreading of the text.

Line 63, were eligible (not are eligible)

Response: Thank you. We have made the correction.

Line 172, “after adjustment for the number of chronic diseases” (not adjusted to)

Response: Thank you. We have made the correction.

Line 176, was present in 54.9% (not presented)

Response: Thank you. We have made the correction.

Line 197, the CFS is gaining popularity (not gaining ITS popularity)

Response: Thank you. We have made the correction.

Line 200, junior medical what? There seems to be a word missing (staff?)

Response: Thank you. We have made the correction.

Line 204, the CFS also had good agreement (not have)

Response: Thank you. We have made the correction.

Line 228, should not be generalised to all older patients

Response: Thank you. We have made the correction.